# Portable Waveguide-Based Optical Biosensor

**DOI:** 10.3390/bios12040195

**Published:** 2022-03-25

**Authors:** Philip A. Kocheril, Kiersten D. Lenz, David D. L. Mascareñas, John E. Morales-Garcia, Aaron S. Anderson, Harshini Mukundan

**Affiliations:** 1Physical Chemistry and Applied Spectroscopy Group, Chemistry Division, Los Alamos National Laboratory, Los Alamos, NM 87545, USA; pkocheril@lanl.gov (P.A.K.); kiersten@lanl.gov (K.D.L.); aaronsa@lanl.gov (A.S.A.); 2National Security Education Center, Los Alamos National Laboratory, Los Alamos, NM 87545, USA; dmascare@gmail.com (D.D.L.M.); jevan.morales@gmail.com (J.E.M.-G.)

**Keywords:** biosensor, portable, fluorescence, waveguide, evanescent field

## Abstract

Rapid, on-site diagnostics allow for timely intervention and response for warfighter support, environmental monitoring, and global health needs. Portable optical biosensors are being widely pursued as a means of achieving fieldable biosensing due to the potential speed and accuracy of optical detection. We recently developed the portable engineered analytic sensor with automated sampling (PEGASUS) with the goal of developing a fieldable, generalizable biosensing platform. Here, we detail the development of PEGASUS’s sensing hardware and use a test-bed system of identical sensing hardware and software to demonstrate detection of a fluorescent conjugate at 1 nM through biotin-streptavidin chemistry.

## 1. Introduction

Point-of-care diagnostics are essential for early community intervention during outbreaks and to guide treatment decisions for various diseases. The term “point-of-care” describes all diagnostic tests that can be performed as close as possible to the patient, providing analytical results in a very short period for an immediate diagnostic or therapeutic decision. Point-of-care biomarker diagnostics require (1) the identification of critical, specific biomarkers, (2) the development of assays for their measurement in complex biological samples like sputum, blood, urine, and cerebrospinal fluid, and (3) rapid, sensitive, specific, quantitative, integrated, and portable biosensor platforms that are compatible with those assays [1,2,3,4]. Over the past few decades, the discovery of biomarkers and the development of biomarker-based diagnostic assays have progressed significantly, becoming clinically relevant for conditions such as cancer and infectious diseases [5,6,7,8,9,10]. However, the ability to rapidly and quantitatively measure specific biological signatures at the point of need remains challenging.

Biosensor platforms have evolved alongside biomarker-based diagnostic assays [10,11]. Portable biosensors are sought as a deployable means to monitor water quality, measure environmental pollution, assess warfighter health, detect pathogens in a point-of-care setting, and more [12,13,14,15,16,17,18]. Techniques such as interferometry, microwave sensing, surface plasmon resonance, fluorimetry, and Bloch surface wave sensing have been applied to the development of portable biosensing technologies [19,20,21,22,23,24,25,26,27]. In particular, fluorescent waveguide-based biosensors hold significant promise in portable applications due to their small size, potential low cost, relative ease of use, low (theoretically zero) background signal, and fast, low-noise detection with silicon photodiodes [28,29].

Our team at Los Alamos National Laboratory has previously reported a benchtop waveguide-based optical biosensor (WOB) that combines the spatial specificity of evanescent field sensing, the specificity of biotin-streptavidin binding, and the spectral sensitivity of a fluorescence detection platform [30]. This biosensor has been used to detect many different compounds of biochemical interest, including lipopolysaccharides, lipoteichoic acids, lipoarabinomannan, protein toxins, disease biomarkers, and viral nucleic acids [31,32,33,34,35,36]. In addition, we previously developed a highly portable biosensor, but that platform was designed to specifically detect cholera toxin and ricin [37,38]. For widespread use in point-of-care settings, it is important for a biosensor to be amenable to a wide array of targets. To address this requirement, we developed the portable engineered analytic sensor with automated sampling (PEGASUS) [39], which we present here.

PEGASUS was designed to miniaturize and integrate the sensing ability of WOB with a microfluidic chip for sample processing toward the goal of developing a truly fieldable biosensor. The details of our microfluidic platform have been described elsewhere [40]. PEGASUS additionally differs from WOB because it uses a different, smaller waveguide mounting apparatus. Light-coupling efficiency and overall sensing performance can vary between different waveguides. Therefore, to directly compare the performance of PEGASUS and WOB, we built a PEGASUS test-bed sensor (PTB) with identical sensing hardware and software to PEGASUS. PTB uses the same waveguide mounting system as WOB for cross-compatibility while maintaining portability, fitting into a 22″ × 14″ × 9″ case and weighing 19.4 lbs. Here, we present the design of PEGASUS’s biosensing hardware and evaluate the performance of PEGASUS’s sensing hardware and software through a comparison between PTB and WOB.

## 2. Experimental

### 2.1. Sensor Design

The design of WOB (Figure 1a) has been described previously [30,38]. In brief, 532-nm light from a frequency-doubled diode-pumped laser (GCL-025-S, CrystaLaser LC, Reno, NV, USA; 25 mW; laser head: 50 mm × 36 mm × 120 mm, 1.3 lbs; CrystaLaser CL-2000 power supply: 50 mm × 140 mm × 150 mm, 1 lb) is passed through a variable attenuator consisting of a neutral density (ND) filter (1 stop), a polarizing filter, and a zero-order half-wave plate, attenuating the laser’s total power to ≈520 μW to minimize photobleaching during data acquisition. The beam is passed through a digitally driven mechanical shutter (Model 845HP, Newport Corporation, Irvine, CA, USA; shutter head: 57.2 mm × 34.3 mm × 29 mm; controller: 82.5 mm × 152.5 mm × 159 mm; total weight 3.5 lbs) and focused (f = 200 mm; beam waist ≈1 mm) onto the diffraction grating of a silicon oxynitride single-mode planar optical waveguide (nGimat Ltd., Atlanta, GA, USA; Spectrum Thin Films Inc., Hauppauge, NY, USA). The beam of the laser can be thought of as being split into three main components at the waveguide–air interface: (1) light that is reflected off the surface of the waveguide, (2) light that is coupled into the thin film of the waveguide, and (3) light that is transmitted through the waveguide. The light that is not reflected off the surface of the waveguide (≈80 μW) or coupled into the waveguide (≈30–400 μW depending on the waveguide) is transmitted through the waveguide and monitored with a silicon photodiode power meter (S120C, Thorlabs, Newton, NJ, USA). The evanescent field generated by the light coupled into the waveguide excites fluorophores held near the surface of the waveguide. Isotropically emitted fluorescence is collected with a fiber-optic cable (QP600-025-UV-BX, Ocean Insight, Orlando, FL, USA; held roughly normal at 1 mm away from the surface of the waveguide). The collected light is routed through a 532-nm long-pass filter and coupled into a fiber-optic spectrometer (P600-2-UV-VIS, Ocean Insight; OceanOptics USB2000, Ocean Insight). The resulting data is transmitted to a computer (Dell Latitude D520; 5.9 lbs) running a LabVIEW-based Virtual Instrument that coordinates the shutter control with data acquisition and processes spectra (SCB-68, National Instruments, Austin, TX, USA; LabVIEW v7.1, National Instruments; OmniDriver, Ocean Insight).

We designed PEGASUS and PTB (Figure 1b) with the goal of miniaturizing our previous sensor by removing or replacing large components whenever possible. PTB is equipped with a compact diode laser (CPS532-C2, Thorlabs; 900 μW; cylindrical, 11 mm × 72.8 mm; 0.08 lbs) mounted in a Thorlabs 30-mm optical cage system, removing the need for the variable attenuator by using a weaker laser. The laser is focused (f = 125 mm; beam waist ≈1 mm) onto the diffraction grating of a silicon oxynitride single-mode planar optical waveguide (≈100 μW reflected, ≈30–170 μW coupled) and monitored with a power meter (S120C, Thorlabs). Isotropically emitted fluorescence from the thin film of the waveguide is collected with a fiber-optic cable (FG550UEC, Thorlabs; held roughly normal at ≈1 mm away from the surface of the waveguide), passed through a 532-nm long-pass filter, and coupled into a fiber-optic spectrometer (QP600-025-UV-BX, Ocean Insight; OceanOptics Flame-S, Ocean Insight). A Raspberry Pi (RPi; Raspberry Pi 3 Model B v1.2, Raspberry Pi Foundation, Cambridge, UK; 0.25 lbs) provides controllable power to the laser, thereby eliminating the need for the shutter. We use custom-written Python code (Tkinter; OceanOptics SeaBreeze, Ocean Insight) to process spectra on the RPi, replacing the laptop with a more compact and power-efficient computer. Alternatively, emulators (e.g., Box86) enable running x86 spectral processing software packages (e.g., OceanView v2.0.8, Ocean Insight) on an RPi.

As shown in Figure 1c, PTB fits in a 22″ × 14″ × 9″ case (Model 1535, Pelican, Torrance, CA, USA; 8.7 lbs) equipped with a monitor, keyboard, and mouse. Including the weight of the case, this contained sensing platform weighs 19.4 lbs. Because the RPi provides power to both the laser and the spectrometer and draws minimal power itself, the sensor hardware can be powered by a battery (at least 5 V, 2.5 A is required). Equipped with a USB-powered monitor, data acquisition can be performed entirely without mains electrical power. Alternatively, the power requirements of the system are small enough that a portable power station (e.g., Explorer 160, Jackery Inc., Fremont, CA, USA) is more than sufficient to power the sensor and peripherals that use mains power.

### 2.2. Methods

#### 2.2.1. Flow Cell Preparation

We prepared our flow cell using previously published procedures [30,33,34,35,36,41,42]. A planar optical waveguide and a glass coverslip (3″ × 1″ glass microscope slide with two 1-mm holes drilled 1.5 cm from the center along the long axis of the coverslip; 48300-036, VWR International, Radnor, PA, USA) were cleaned for 5 min each in chloroform (319988, Millipore Sigma, St. Louis, MO, USA), ethanol (EX0276, Millipore Sigma), and ultrapure water (Direct-Q 3 UV-R, Millipore Sigma) by bath sonication (2510R-DTH Ultrasonic Cleaner, Branson Ultrasonics, Brookfield, CT, USA), dried under argon gas (Airgas, Radnor, PA, USA), and further cleaned by ultraviolet-ozone treatment (Model T10 × 10/OES, UVOCS Inc., Lansdale, PA, USA) for 40 min. The cleaned waveguide and coverslip were bonded together with a hydrophobic gasket (laser-cut 3″ × 1″ silicone sheet with a 1.5-mm-radius and 3-cm-straight side length geometric stadium cut out of the center of the gasket; CultureWell Silicone Sheet Material RD477403-M, Grace Bio-Labs, Bend, OR, USA) to form a ≈60-μL flow cell. The waveguide–gasket–coverslip assembly was clamped between two pieces of a custom-milled housing fitted with an O-ring-sealed septum (inlet) and an O-ring-sealed drain tube (outlet) that align with the holes on the coverslip.

#### 2.2.2. Lipid Preparation

Supported lipid bilayers were prepared using previously published procedures [30,33,34,35,36,41,42]. 60 μL of 5 mM 1,2-dioleoyl-sn-glycero-3-phosphocholine (850345, Avanti Polar Lipids, Alabaster, AL, USA) in chloroform and 0.6 μL of 5 mM 1,2-dioleoyl-sn-glycero-3-phosphoethanolamine-*N*-(cap biotinyl) (870273, Avanti Polar Lipids) in chloroform were deposited in a glass test tube by syringe, dried under a gentle stream of argon gas, and reconstituted in 600 µL of filter-sterilized Dulbecco’s phosphate-buffered saline (PBS; D8662, Millipore Sigma) for a final total lipid concentration of 0.5 mM. The lipids were shaken for 30 min at room temperature (≈120 RPM) and passed ten times unidirectionally through the 0.1-μm polycarbonate membrane of a lipid extruder (Mini-Extruder 610000, Avanti Polar Lipids) at room temperature to prepare unilamellar vesicles (final volume ≈590 μL). 70 µL of prepared lipids were pipetted into the assembled flow cell, sealed (ST200 Adhesive Seal Tabs, Grace Bio-Labs), and incubated at 4 °C overnight (≈16 h) to allow fusion of a bilayer to the surface.

#### 2.2.3. Fluorescence Assays

The flow cell was washed with 2 mL of PBS (flow rate ≈10 mL/min) and 2 mL of 0.5% bovine serum albumin (BSA; A7906, Millipore Sigma) in PBS (≈10 mL/min). Both biosensors were aligned by maximizing the intensity of the streak resulting from total internal reflection in the thin film of the waveguide (WOB: 391 μW coupled; PTB: 62 μW coupled). The lipid bilayer on the waveguide was blocked for 1 h at room temperature with 2 mL of 2% BSA in PBS (≈10 mL/min) and washed with 2 mL of 0.5% BSA in PBS (≈10 mL/min). Five background spectra were recorded on each sensor. All spectra were recorded from 400–700 nm in a dark room at room temperature with a black box placed on top of the sensor, an integration time of 3 s, and a ± 3 unweighted moving window average. PTB spectra were recorded as the average of three scans. The waveguide was incubated for 5 min at room temperature with a 250-μL injection of 1 nM streptavidin-Alexa Fluor 532 conjugate (SA-AF532; S11224, Thermo Fisher Scientific, Waltham, MA, USA) in PBS (≈1.2 mL/min). The cell was washed with 2 mL of 0.5% BSA in PBS (≈10 mL/min).

A fluorescence spectrum was recorded on WOB. The assembled flow cell was quickly moved to PTB and a fluorescence spectrum was recorded on PTB, providing a matched pair of spectra suitable for direct comparison. The SA-AF532 on the waveguide was photobleached by exposure to 532-nm laser light for 10 min on WOB, and a new background spectrum was recorded on each sensor following photobleaching. A total of four spectra were acquired for each sensor by performing this procedure three additional times with the same waveguide and lipid bilayer, alternating the order in which the sensors were used. Spectra were background-corrected by subtracting the most recently acquired background spectrum from the observed fluorescence spectrum.

## 3. Results and Discussion

We designed PEGASUS, a miniaturized version of WOB, with the goal of developing a generalizable biosensor that could meet the distinct need for sensitive, portable, and rapid biosensing. We have developed, optimized, and validated assays for several emerging biological challenges on WOB [31,32,33,34,35,36]. Because PEGASUS and WOB use similar biological assay architectures and general methodologies, PEGASUS is compatible with the lipid, protein, and nucleic acid sensing assays that we have previously described [31,32,33,34,35,36].

We have previously demonstrated that several functional surfaces are compatible with our biosensing platforms [41,43]. In this early validation study, we used a supported biotinylated lipid bilayer for its simplicity, ease of use, and nearly quantitative binding interaction with streptavidin (K_d_ = 40 fM) and its commercially available fluorescent conjugates [44]. Representative fluorescence spectra of 1 nM SA-AF532 taken on WOB and PTB are shown in Figure 2. Both WOB and PTB are clearly capable of detecting evanescent field-stimulated fluorescence at ≈560 nm from a low-nanomolar analyte.

Because PTB and WOB use different spectrometers with different scales of relative fluorescent intensity, we instead use the average signal-to-noise ratio (SNR) as a generalizable metric to compare sensor performance (Figure 3a). As shown in Figure 3b, PTB (SNR ≈ 3.4 ± 1.6) exhibits a smaller estimated SNR than WOB (SNR ≈ 16.3 ± 4.8). The largest contributor to the difference in SNRs between WOB and PTB is likely the difference in coupling efficiency between the sensors. Defined as the percentage of incident, non-reflected light that is coupled into the waveguide (as opposed to the light that is transmitted through the waveguide), WOB exhibits an estimated coupling efficiency of ≈89% and PTB exhibits an estimated coupling efficiency of ≈8% in these experiments. A detailed discussion of the intricacies of polarization, refraction, and reflection is beyond the scope of this manuscript [45], but we do note that the additional optics used in WOB are likely responsible for the significant difference in coupling efficiency between WOB and PTB.

Although PTB exhibits a smaller estimated SNR than WOB, diminished sensitivity is an oft-encountered challenge in the development of miniaturized and fieldable sensors [46]. Furthermore, WOB is consistently more sensitive than conventional immunoassays using identical assay architectures (often by multiple orders of magnitude) [31,34]. Therefore, given that the SNR of PTB is within a factor of five of that of WOB, PEGASUS is likely more than sufficient for use with our biosensing assays, which generally employ bright fluorophores with high excitation efficiencies and quantum yields [31,32,33,34,35,36]. Additionally, further optimizations such as metal-enhanced fluorophores and other surface functionalizations could be employed in the future to improve the sensitivity of this biosensing platform [10,47,48,49]. Ultimately, the advantage of the portability of this platform outweighs the disadvantage of reduced sensitivity.

## 4. Conclusions

Integrated and portable biosensors in a point-of-care format enable rapid, on-site analytical measurements, making many use-cases possible that would not be possible with benchtop laboratory instrumentation (e.g., measuring water samples at a riverside, assessing warfighter biofluid samples when deployed, and analyzing patient biofluid samples in a point-of-care setting). We have developed a portable waveguide-based optical biosensor (PEGASUS) and used a test-bed (PTB) of identical sensing hardware and software to evaluate the sensor’s performance. PTB fits into a 22″ × 14″ × 9″ case and weighs 19.4 lbs. The creation and validation of this sensor is an important step toward a truly fieldable biosensor platform. Future work will focus on increasing the sensitivity of this platform and exploring performance in complex samples under field conditions.

## Figures and Tables

**Figure 1 biosensors-12-00195-f001:**
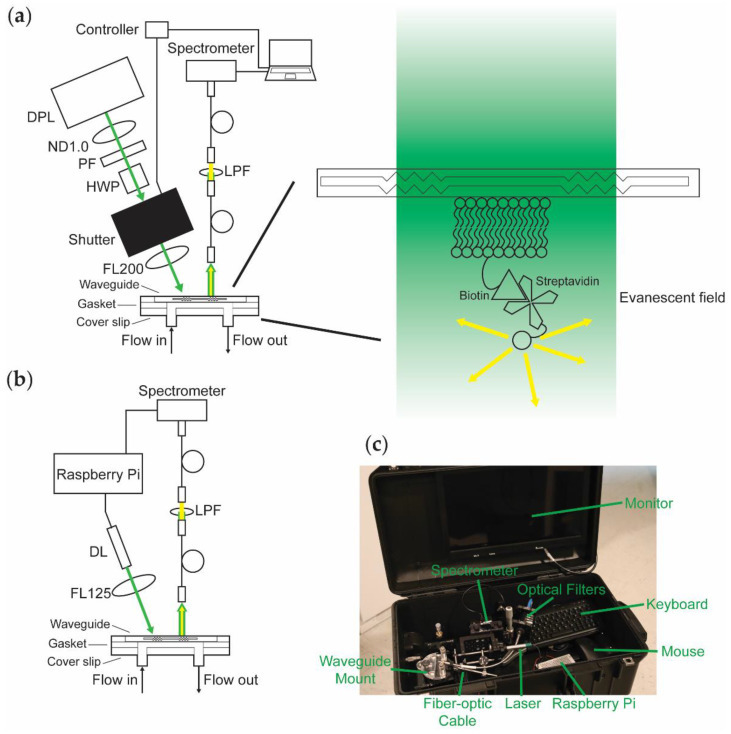
Top-down comparison of the biosensor platforms being evaluated. (**a**) WOB diagram. DPL: diode-pumped laser; ND1.0: 1-stop neutral density filter; PF: polarizing filter; HWP: half-wave plate; FL200: focusing lens with 200-mm focal length; LPF: long-pass filter. (**b**) PTB diagram. DL: diode laser; FL125: focusing lens with 125-mm focal length. (**c**) Photograph of PTB inside of a 22″ × 14″ × 9″ case equipped with a monitor (embedded in the top panel of the case), keyboard, and mouse (total weight 19.4 lbs).

**Figure 2 biosensors-12-00195-f002:**
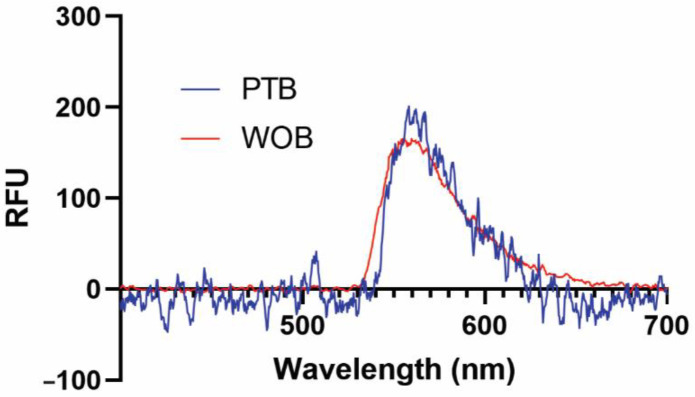
Representative fluorescence spectra of 1 nM streptavidin-Alexa Fluor 532 conjugate from 400–700 nm recorded on WOB (red) and PTB (blue). RFU: relative fluorescence units.

**Figure 3 biosensors-12-00195-f003:**
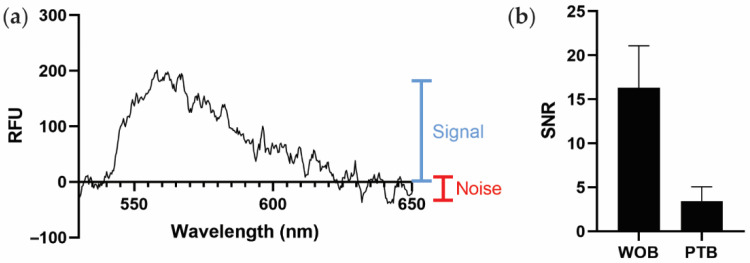
Comparison of biosensor performance. (**a**) Graphical representation of signal (blue) and noise (red) on a spectrum of PTB recorded with 1 nM SA-AF532, as used to estimate the signal-to-noise ratio (SNR) in this work. RFU: relative fluorescence units. (**b**) Comparison of average SNRs of WOB and PTB. Error bars are ± one standard deviation (*n* = 4).

## Data Availability

Spectral data are publicly available on Figshare (DOI: 10.6084/m9.figshare.17955254).

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
