# Peer review of "Portable Waveguide-Based Optical Biosensor"

_biosensors, 2022, doi:10.3390/bios12040195_

Round 1
Reviewer 1 Report
The manuscript entitled ‘Portable waveguide-based optical biosensor’ proposes a new platform called PEGASUS for the portable waveguide optical detection of biomolecules (protein and nucleic acids) in PoC applications. The work is interesting and the level of technological integration in PEGASUS is promising for the desired applications, however there are some major revisions to be done before considering it for publication in this journal.
- The introduction section is too small and poor of contents. Examples from literature reporting the state-of-the-art of waveguide-based optical biosensors and their applications in PoCs are missing. Please revise including a critical review of the current optical sensing technologies focusing on the advantages and limits of their use in a PoC format.
- There are some points that need be checked:
- Line 23-27: “Point-of-care diagnostics require (1) the identification of critical, specific biomarkers, (2) the development of assays for their measurement in complex biological samples like sputum, blood, urine, and cerebrospinal fluid, and (3) rapid, sensitive, specific, quantitative, and portable biosensor platforms that are compatible with those assays”. The sentence is difficult to understand. The term point-of-care (PoC) refers to all diagnostic tests that can be performed as close as possible to the patient, providing analytical results in a very short period for an immediate diagnosis and/or therapeutic decision. To this purpose PoCs need portable, integrated and user-friendly solution to be used outside a laboratory by unspecialized personnel. Please revise reporting a better description of PoCs.
- Line 36: the term “fluorescence” doesn’t mean a technique itself. I would specify better the type of method (e.g. fluorescence microscopy, imaging, etc.).
- Line 91: Figure 1c is not intuitive. I suggest to include some captions describing elements inside the case.
- Line 144-148: what is the lipid final concentration obtained? Is there any loss of material during the sample preparation? I would revise including this specification.
- Line 197: why is the PTB signal so noised compared to the WOB one? Measurements of a range of SA-AF532 concentrations (above and below 1 nM) would help to define the Limit of Detection (LoD) and Quantification (LoQ) of PTB.
- Line 199-201: the sentence is not clear to me. Each fluorescence acquisition requires a signal normalization in order to exclude any alteration given by the experimental and reagents conditions. Please revise explaining better this statement.
- Line 213: which concentration of SA-AF532 has been used to perform the test reported in Figure 3? Please include this specification.
- Line 220-222: literature reports evidences of stable and enhanced fluorophores and surface functionalizations for highly sensitive optical biosensing applications that here are not mentioned: "Jeong Y, Kook YM, Lee K, Koh WG. Metal enhanced fluorescence (MEF) for biosensors: General approaches and a review of recent developments. Biosens Bioelectron. 2018 Jul 15;111:102-116. doi: 10.1016/j.bios.2018.04.007. Epub 2018 Apr 7. PMID: 29660581"; "L. Sciuto, M.F. Santangelo, G. Villaggio, F. Sinatra, C. Bongiorno, G. Nicotra, S. Libertino, Photo-physical characterization of fluorophore Ru(bpy)32+ for optical biosensing applications, Sensing and Bio-Sensing Research 6, 2015, 67-71" ; "Mukundan H, Anderson AS, Grace WK, Grace KM, Hartman N, Martinez JS, Swanson BI. Waveguide-based biosensors for pathogen detection. Sensors (Basel). 2009;9(7):5783-809"; "Sciuto EL, Bongiorno C, Scandurra A, Petralia S, Cosentino T, Conoci S, Sinatra F, Libertino S. Functionalization of Bulk SiO2 Surface with Biomolecules for Sensing Applications: Structural and Functional Characterizations. Chemosensors. 2018; 6(4):59". I would include these evidences, highlighting the importance of using such stable dyes and efficient surface modifications in order to avoid SNR, sensitivity and specificity common issues of optical detection.
- A complete evaluation of the platform analytical performances is missing. In my opinion, going deeper in detail with the experiments, reporting a complete characterization of the detection module (PTB/PEGASUS), a sensing calibration at various concentration of strepatividine-biotinylated lipid bilayer complex and the relative LoD/LoQ extimation, is necessary to prove the sensor suitability for PoC applications.
- A direct sensing tests with the PEGASUS platform, and not its test-bed analogue, would increase the strength of work, especially using more significant molecular targets for on-site diagnostics than streptavidin/biotin complex.
After refinement and completion, the manuscript could be a good advancement in this topic.
Author Response
Dear reviewer
Thank you for your clear and thorough review of the manuscript, and constructive feedback. we have attempted to address all your concerns in the comments below, and the changes have been incorporated into revised manuscript. We trust you will find the manuscript now suited for publication in Biosensors.
Sincerely
Harshini Mukundan, Corresponding author
Reviewer 1
Comment: The introduction section is too small and poor of contents. Examples from literature reporting the state-of-the-art of waveguide-based optical biosensors and their applications in PoCs are missing. Please revise including a critical review of the current optical sensing technologies focusing on the advantages and limits of their use in a PoC format.
Response: We thank the reviewer for their feedback. We have edited the manuscript in order to expand on the introduction section, and have also incorporated additional review articles in order to strengthen the same (Chen & Wang 2020; Ligler et al. 2007). These references provide further background information on the current state of optical biosensing and portable waveguide sensing technologies, in addition to the biosensor reviews included in our initial submission (Benito-Pena et al. 2016, Mukundan et al. 2009, etc.). Beyond this, most of the reviews t in the literature either focus on a specific target/disease, or describe specific biosensing technology. We have included a couple of these reviews in our submission as well (Eyvazi et al. 2021, Chocarro-Ruiz et al. 2017, etc.). In the context of fluorescent waveguide biosensors, the review by Benito-Pena et al. that we referenced in our initial submission is comprehensive, and presents a comparison of a variety of sensing platforms and technologies. In addition, Mukundan et al 2009 is a comprehensive summarization of waveguide based biosensor platforms as well.
Comment: Line 23-27: “Point-of-care diagnostics require (1) the identification of critical, specific biomarkers, (2) the development of assays for their measurement in complex biological samples like sputum, blood, urine, and cerebrospinal fluid, and (3) rapid, sensitive, specific, quantitative, and portable biosensor platforms that are compatible with those assays”. The sentence is difficult to understand. The term point-of-care (PoC) refers to all diagnostic tests that can be performed as close as possible to the patient, providing analytical results in a very short period for an immediate diagnosis and/or therapeutic decision. To this purpose PoCs need portable, integrated and user-friendly solution to be used outside a laboratory by unspecialized personnel. Please revise reporting a better description of PoCs.
Response: We have now clarified our language regarding point-of-care diagnostics. We first use the reviewer’s concise description as a general definition of point-of-care diagnostics, then specify how the requirements that we outlined apply to biomarker diagnostics designed to be used in a point-of-care setting.
Old text (lines 23-24): Point-of-care diagnostics require…
Revised text (lines 23-26 in the revised manuscript): The term “point-of-care” describes all diagnostic tests that can be performed as close as possible to the patient, providing analytical results in a very short period for an immediate diagnostic or therapeutic decision. Point-of-care biomarker diagnostics require…
Comment: Line 36: the term “fluorescence” doesn’t mean a technique itself. I would specify better the type of method (e.g. fluorescence microscopy, imaging, etc.).
Response: We have updated our language to specifically describe fluorimetry, or the measurement of fluorescent intensity, as the associated technique rather than the general process of fluorescence.
Old text (lines 35-36 in the initial submission): Techniques such as interferometry, microwave sensing, surface plasmon resonance, and fluorescence have been applied…
Revised text (lines 38-40 in the revised manuscript): Techniques such as interferometry, microwave sensing, surface plasmon resonance, fluorimetry, and Bloch surface wave sensing have been applied…
Comment: Line 91: Figure 1c is not intuitive. I suggest to include some captions describing elements inside the case.
Response: We have added labels to the key parts of the image in Figure 1c to clarify the components present.
Old image (line 90):
New image (line 98 in the revised manuscript):
Comment: Line 144-148: what is the lipid final concentration obtained? Is there any loss of material during the sample preparation? I would revise including this specification.
Response: The final lipid concentration obtained is 0.5 mM. The lipids are prepared in excess of the saturating concentration to form a bilayer on the waveguide, so any loss of material due to sticking in the tube is negligible. There is a slight loss of volume through extrusion, but the losses are again negligible because the lipids are prepared in excess of what will be needed for multiple experiments. We have added a few words to clarify these points.
Old text (lines 148-149, 151-152 in the initial submission): … 600 µL of filter-sterilized Dulbecco’s phosphate-buffered saline (PBS; D8662, Millipore Sigma). The lipids were shaken… to prepare unilamellar vesicles. 70 µL of prepared lipids…
Revised text (lines 156-158, 160-161 in the revised manuscript): … 600 µL of filter-sterilized Dulbecco’s phosphate-buffered saline (PBS; D8662, Millipore Sigma) for a final total lipid concentration of 0.5 mM. The lipids were shaken… to prepare unilamellar vesicles (final volume 590 μL). 70 µL of prepared lipids…
Comment: Line 197: why is the PTB signal so noised compared to the WOB one? Measurements of a range of SA-AF532 concentrations (above and below 1 nM) would help to define the Limit of Detection (LoD) and Quantification (LoQ) of PTB.
Response: The coupling efficiency of PTB is not as effective as that of the WOB. This is likely due to the additional polarizing optics present in WOB. We attribute the increased noise observed in PTB to this diminished overall coupling efficiency. We eliminated the polarizing optics in order to simplify the PTB, and enhance its portability. This has been included in the revised manuscript as below.
Revised text (lines 205-207 in the revised manuscript): WOB contains polarizing optics, whereas PTB does not. This is the likely cause of the increased noisiness of the PTB spectra, over the WOB one.
We agree that measurements of a range of concentrations would help to define a limit of detection for PTB. However, given that the purpose of this manuscript was to demonstrate the engineering design of the PTB, and very preliminary functionality via the measurement of streptavidin, we did not perform said measurements. Another reason for this is the very high binding affinity of biotin to streptavidin. Indeed, We propose to perform limit of detection assessments upon application of this biosensor to a biologically relevant analyte. We concur that limit of detection – particularly in a complex biofluid sample – is an incredibly valuable quantity in demonstrating the utility of a sensor for diagnostics applications. In fact, we have previously performed similar characterization many times with WOB for biologically relevant analytes (e.g., Kubicek-Sutherland et al. 2019, Mukundan et al. 2012, etc.). We are currently working on such evaluations for a subsequent, more detailed validation manuscript. We have included a description of this in the conclusion section as below.
Revised text (lines 247-249 in the revised manuscript):We are currently working on validating assay performance, and comparing sensitivity of detection and limit of detection in order to evaluate feasibility for real-world diagnostic applications.
Comment: Line 199-201: the sentence is not clear to me. Each fluorescence acquisition requires a signal normalization in order to exclude any alteration given by the experimental and reagents conditions. Please revise explaining better this statement.
Response: We apologize for the lack of clarity in this sentence. Normalization is traditionally an essential step in the comparison of multiple fluorescence spectra. However, because we acquired spectra in matched pairs by moving our assembled waveguide flow cell between biosensors, normalization is unnecessary. We see that the language we used was unclear, as it was intended to be a general comment on the comparability of fluorimetry assays. We have now replaced the general comment with a specific note on the comparability of these two biosensors.
Old text (lines 199-201 in the initial submission): Because the observed signal of a fluorescence assay will depend on the specific conditions and reagents used in the assay, we instead use the average signal-to-noise ratio (SNR)…
Revised text (lines 208-211 in the revised manuscript): Because PTB and WOB use different spectrometers with different scales of relative fluorescent intensity, we instead use the average signal-to-noise ratio (SNR)…
Comment: Line 213: which concentration of SA-AF532 has been used to perform the test reported in Figure 3? Please include this specification.
Response: The caption of Figure 3 has been updated to include the concentration of SA-AF532 (1 nM) used to acquire the spectrum.
Old text (lines 213-214 in the initial submission): Graphical representation of signal (blue) and noise (red), as used to estimate the signal-to-noise ratio (SNR) in this work.
Revised text (lines 223-225 in the revised manuscript): Graphical representation of signal (blue) and noise (red) on a spectrum of PTB recorded with 1 nM SA-AF532, as used to estimate the signal-to-noise ratio (SNR) in this work.
Comment: Line 220-222: literature reports evidences of stable and enhanced fluorophores and surface functionalizations for highly sensitive optical biosensing applications that here are not mentioned: "Jeong Y, Kook YM, Lee K, Koh WG. Metal enhanced fluorescence (MEF) for biosensors: General approaches and a review of recent developments. Biosens Bioelectron. 2018 Jul 15;111:102-116. doi: 10.1016/j.bios.2018.04.007. Epub 2018 Apr 7. PMID: 29660581"; "L. Sciuto, M.F. Santangelo, G. Villaggio, F. Sinatra, C. Bongiorno, G. Nicotra, S. Libertino, Photo-physical characterization of fluorophore Ru(bpy)32+ for optical biosensing applications, Sensing and Bio-Sensing Research 6, 2015, 67-71" ; "Mukundan H, Anderson AS, Grace WK, Grace KM, Hartman N, Martinez JS, Swanson BI. Waveguide-based biosensors for pathogen detection. Sensors (Basel). 2009;9(7):5783-809"; "Sciuto EL, Bongiorno C, Scandurra A, Petralia S, Cosentino T, Conoci S, Sinatra F, Libertino S. Functionalization of Bulk SiO2 Surface with Biomolecules for Sensing Applications: Structural and Functional Characterizations. Chemosensors. 2018; 6(4):59". I would include these evidences, highlighting the importance of using such stable dyes and efficient surface modifications in order to avoid SNR, sensitivity and specificity common issues of optical detection.
Response: These references have been added where specified, along with the context that other enhanced fluorophores and surface functionalization techniques could be used to further improve the sensitivity of this biosensing platform.
Old text (lines 220-224 in the initial submission): Therefore, PTB is more than sufficient for use with our biosensing assays, which generally employ bright fluorophores with high excitation efficiencies and quantum yields [27-32], and the advantage of the portability of this platform outweighs the disadvantage of reduced sensitivity.
Revised text (lines 230-236 in the revised manuscript): Therefore, PTB is more than sufficient for use with our biosensing assays, which generally employ bright fluorophores with high excitation efficiencies and quantum yields [29-34]. Additionally, further optimizations such as metal-enhanced fluorophores and other surface functionalizations could be employed in the future to improve the sensitivity of this biosensing platform [10,45-47]. Ultimately, the advantage of the portability of this platform outweighs the disadvantage of reduced sensitivity.
Comment: A complete evaluation of the platform analytical performances is missing. In my opinion, going deeper in detail with the experiments, reporting a complete characterization of the detection module (PTB/PEGASUS), a sensing calibration at various concentration of strepatividin-biotinylated lipid bilayer complex and the relative LoD/LoQ extimation, is necessary to prove the sensor suitability for PoC applications.
Response: The purpose of this first concise communication was intended to demonstrate the engineering features, and preliminary functionality of the biosensor platform. The biosensor’s hardware is not a limiting factor in the analytical sensitivity of the sensor. In addition, we have ensured that the sensing components of the well-validated WOB are not compromised or altered in the engineering design of the PTB, which is what we present in this first short paper. We will be following this up with a full article on comparative assay validation and associated metrics. In fact, we have been working on proteins, nucleic acid, and lipid detection on this interface, and this will be covered in future articles.
Comment: A direct sensing tests with the PEGASUS platform, and not its test-bed analogue, would increase the strength of work, especially using more significant molecular targets for on-site diagnostics than streptavidin/biotin complex.
Response: We agree that a direct comparison between PEGASUS and WOB would be ideal rather than using PTB as an analogue of PEGASUS. However, as we noted, the waveguide mounting systems used by PEGASUS and WOB are not interchangeable. This would bias the initial comparisons, which is why we developed the PTB platform. The comparison between biosensors that we present here is only valid because the waveguide, flow cell, lipids, etc. were identical and moved between biosensors. We propose to introduce the actual PEGASUS platform, and perform a three way comparison in the next validation manuscript.

Reviewer 2 Report
In the present work, Philip A. Kocheril and co-authors developed a compact optical biosensor based on waveguide by starting from an identical system previously developed. The authors provide a sufficiently clear description of the hardware and software systems used to realize the integrated biosensor and the work is completed by an experimental demonstration of the potential application of the assembled biosensor. In the present work, the author compared the performances of the two systems by detecting a fluorescent conjugated through biotin-streptavidin chemistry. Although the integrated system shows a larger signal-to-noise ratio, the authors claim that it is enough sensitive to detect 1 nM of streptavidin Alexa Fluor 532 conjugate. The presented results are clear and support the author’s conclusions.
Apart from a few comments below, I consider the manuscript suitable for publication in this journal.
- In the introduction of the manuscript, the authors did a list of optical techniques which are suitable for making integrated sensors with enough sensitivity. Recently, Bloch surface waves-based sensors have been demonstrated as a valid alternative technology for label-free and fluorescence sensing, therefore, I suggest to the authors cite the following for completeness: “K. Toma, E. Descrovi, M. Toma, M. Ballarini, P. Mandracci, F. Giorgis, A. Mateescu, U. Jonas, W. Knolla, and J. Dostálek, Bloch surface wave-enhanced fluorescence biosensor, Biosens. Bioelectron., vol. 43, pp. 108-114, 2013” and/or “Occhicone, A.; Del Porto, P.; Danz, N.; Munzert, P.; Sinibaldi, A.; Michelotti, F. Enhanced Fluorescence Detection of Interleukin 10 by Means of 1D Photonic Crystals. Crystals 2021, 11, 1517. https://doi.org/10.3390/cryst11121517”.
- Line 77-80. I suggest rephrasing the sentence from line 77 to line 80, in my opinion, it is not clear: is the light divided into reflected one and coupled into the waveguide?
- There is not any description/reference to the inset of Figure 1(a) either into the main text or into the figure caption.
Author Response
Dear Reviewer
Thank you for your detailed and constructive feedback. We have attempted to address all your comments in the description below, and have incorporated them into the manuscript as well, as indicated. We trust you will find the revised manuscript more suitable for publication in Biosensors.
Sincerely and with thanks
Harshini Mukundan, Corresponding Author
Reviewer 2
Comment: In the introduction of the manuscript, the authors did a list of optical techniques which are suitable for making integrated sensors with enough sensitivity. Recently, Bloch surface waves-based sensors have been demonstrated as a valid alternative technology for label-free and fluorescence sensing, therefore, I suggest to the authors cite the following for completeness: “K. Toma, E. Descrovi, M. Toma, M. Ballarini, P. Mandracci, F. Giorgis, A. Mateescu, U. Jonas, W. Knolla, and J. Dostálek, Bloch surface wave-enhanced fluorescence biosensor, Biosens. Bioelectron., vol. 43, pp. 108-114, 2013” and/or “Occhicone, A.; Del Porto, P.; Danz, N.; Munzert, P.; Sinibaldi, A.; Michelotti, F. Enhanced Fluorescence Detection of Interleukin 10 by Means of 1D Photonic Crystals. Crystals 2021, 11, 1517. https://doi.org/10.3390/cryst11121517”.
Response: We thank the reviewer for their feedback. We have added the specified references to the general list of optical techniques that have been applied for biosensing.
Old text (lines 35-37 in the initial submission): Techniques such as interferometry, microwave sensing, surface plasmon resonance, and fluorescence have been applied to the development of portable biosensing technologies [19-23].
Revised text (lines 38-40 in the revised manuscript): Techniques such as interferometry, microwave sensing, surface plasmon resonance, fluorimetry, and Bloch surface wave sensing have been applied to the development of portable biosensing technologies [19-27].
Comment: Line 77-80. I suggest rephrasing the sentence from line 77 to line 80, in my opinion, it is not clear: is the light divided into reflected one and coupled into the waveguide?
Response: We have added an additional sentence to clarify how the laser beam is split at the waveguide-air interface.
Old text: The light that is not reflected off the surface of the waveguide…
Revised text (lines 80-84 in the revised manuscript): The beam of the laser can be thought of as being split into three main components at the waveguide-air interface: (1) light that is reflected off the surface of the waveguide, (2) light that is coupled into the thin film of the waveguide, and (3) light that is transmitted through the waveguide. The light that is not reflected off the surface of the waveguide…
Comment: There is not any description/reference to the inset of Figure 1(a) either into the main text or into the figure caption.
Response: Line 66 of our initial submission (line 69 of the revised manuscript) references Figure 1a specifically, and there is a dedicated section of the caption of Figure 1 (lines 91-93 in the initial submission or 99-101 in the revised manuscript) that describes the components of WOB. We hope that these address the reviewer’s concern in entirety. We have now referenced the Figure 1(a) clearly in these sentences in order to clarify the same.
Reviewer 3 Report
The manuscript deals with built of a portable engineered analytic sensor with automated sampling (PEGASUS test bed - PTB). The authors have described the PEGASUS’s biosensing hardware through a comparison with the Waveguide-based optical biosensor-WOB, previously published. The goal from the authors have been decreased the size of the biosensor, making it miniaturized. The sensor PEGASUS test bed has slight improvement related to the size and signal to noise ratio in relation to the biosensor-WOB, but I can’t see along the text the necessity/applicability to reduce the sensor size. The figure 2, for instance has presented the same result to fluorescence spectra of 1 nM streptavidin-Alexa Fluor with emission peak centered at 560nm to the WOB and PTB biosensor. Therefore, I cannot recommend this paper to publication.
Author Response
Dear Reviewer
Thank you for your detailed and constructive feedback on our manuscript. we have attempted to address all your concerns in entirety and hope that you will find this revised manuscript suitable for acceptance to Biosensors.
Sincerely
Harshini Mukundan, corresponding author
Reviewer 3
Comment: The sensor PEGASUS test bed has slight improvement related to the size and signal to noise ratio in relation to the biosensor-WOB, but I can’t see along the text the necessity/applicability to reduce the sensor size.
Response: We thank the reviewer for their feedback. In addition to the text written in the Introduction of our initial submission that justifies the need for portable biosensors to “monitor water quality, measure environmental pollution, assess warfighter health, detect pathogens, and more” (lines 33-34 in the initial submission), we have now added further text to reiterate the reasons why portable biosensing is valuable in the Conclusion.
Old text (line 226 in the initial submission): We have developed…
New text (lines 239-242 in the revised manuscript): Portable biosensors enable rapid, on-site analytical measurements, making many use-cases possible that would not be possible with benchtop laboratory instrumentation (e.g., measuring water samples at a riverside, assessing warfighter biofluid samples when deployed, and analyzing patient biofluid samples in a point-of-care setting). We have developed…
Comment: The figure 2, for instance has presented the same result to fluorescence spectra of 1 nM streptavidin-Alexa Fluor with emission peak centered at 560nm to the WOB and PTB biosensor.
Response: The similarity of the spectra between WOB and PTB, as shown in Figure 2, is encouraging as the same SA-AF532 system was tested on both biosensors. Therefore, if both biosensors were built and functioning as intended, the spectra should be nearly identical. In this early phase of work, our goal was to demonstrate the engineering efficacy and functionality of the miniaturized platform, which this observation helped us achieve. Additionally, although the PTB spectrum is noisier, the tradeoff between analytical sensitivity and portability is very common for portable biosensor platforms, as noted in the Results and Discussion. We have clarified these concepts in the revised submission, based on your feedback (lines 217-218 in the initial submission, or 227-228 in the revised manuscript).
Round 2
Reviewer 1 Report
Authors provided a revised version of the manuscript where most of the pending issues have been properly solved. There are just few minor points to solve.
- The introduction section is too small and poor of contents. Examples from literature reporting the state-of-the-art of waveguide-based optical biosensors and their applications in PoCs are missing. Please revise including a critical review of the current optical sensing technologies focusing on the advantages and limits of their use in a PoC format.
Response: We thank the reviewer for their feedback. We have edited the manuscript in order to expand on the introduction section, and have also incorporated additional review articles in order to strengthen the same (Chen & Wang 2020; Ligler et al. 2007). These references provide further background information on the current state of optical biosensing and portable waveguide sensing technologies, in addition to the biosensor reviews included in our initial submission (Benito-Pena et al. 2016, Mukundan et al. 2009, etc.). Beyond this, most of the reviews t in the literature either focus on a specific target/disease, or describe specific biosensing technology. We have included a couple of these reviews in our submission as well (Eyvazi et al. 2021, Chocarro-Ruiz et al. 2017, etc.). In the context of fluorescent waveguide biosensors, the review by Benito-Pena et al. that we referenced in our initial submission is comprehensive, and presents a comparison of a variety of sensing platforms and technologies. In addition, Mukundan et al 2009 is a comprehensive summarization of waveguide based biosensor platforms as well.
The introduction section has been expanded and revised as suggested.
- Line 23-27: “Point-of-care diagnostics require (1) the identification of critical, specific biomarkers, (2) the development of assays for their measurement in complex biological samples like sputum, blood, urine, and cerebrospinal fluid, and (3) rapid, sensitive, specific, quantitative, and portable biosensor platforms that are compatible with those assays”. The sentence is difficult to understand. The term point-of-care (PoC) refers to all diagnostic tests that can be performed as close as possible to the patient, providing analytical results in a very short period for an immediate diagnosis and/or therapeutic decision. To this purpose PoCs need portable, integrated and user-friendly solution to be used outside a laboratory by unspecialized personnel. Please revise reporting a better description of PoCs.
Response: We have now clarified our language regarding point-of-care diagnostics. We first use the reviewer’s concise description as a general definition of point-of-care diagnostics, then specify how the requirements that we outlined apply to biomarker diagnostics designed to be used in a point-of-care setting.
Old text (lines 23-24): Point-of-care diagnostics require…
Revised text (lines 23-26 in the revised manuscript): The term “point-of-care” describes all diagnostic tests that can be performed as close as possible to the patient, providing analytical results in a very short period for an immediate diagnostic or therapeutic decision. Point-of-care biomarker diagnostics require…
The PoC description is now clear making easier the text comprehension.
- Line 36: the term “fluorescence” doesn’t mean a technique itself. I would specify better the type of method (e.g. fluorescence microscopy, imaging, etc.).
Response: We have updated our language to specifically describe fluorimetry, or the measurement of fluorescent intensity, as the associated technique rather than the general process of fluorescence.
Old text (lines 35-36 in the initial submission): Techniques such as interferometry, microwave sensing, surface plasmon resonance, and fluorescence have been applied…
Revised text (lines 38-40 in the revised manuscript): Techniques such as interferometry, microwave sensing, surface plasmon resonance, fluorimetry, and Bloch surface wave sensing have been applied…
The term has been revised accordingly.
- Line 91: Figure 1c is not intuitive. I suggest to include some captions describing elements inside the case.
Response: We have added labels to the key parts of the image in Figure 1c to clarify the components present.
Old image (line 90):
New image (line 98 in the revised manuscript):
The figure now reports all descriptions required.
- Line 144-148: what is the lipid final concentration obtained? Is there any loss of material during the sample preparation? I would revise including this specification.
Response: The final lipid concentration obtained is 0.5 mM. The lipids are prepared in excess of the saturating concentration to form a bilayer on the waveguide, so any loss of material due to sticking in the tube is negligible. There is a slight loss of volume through extrusion, but the losses are again negligible because the lipids are prepared in excess of what will be needed for multiple experiments. We have added a few words to clarify these points.
Old text (lines 148-149, 151-152 in the initial submission): … 600 µL of filter-sterilized Dulbecco’s phosphate-buffered saline (PBS; D8662, Millipore Sigma). The lipids were shaken… to prepare unilamellar vesicles. 70 µL of prepared lipids…
Revised text (lines 156-158, 160-161 in the revised manuscript): … 600 µL of filter-sterilized Dulbecco’s phosphate-buffered saline (PBS; D8662, Millipore Sigma) for a final total lipid concentration of 0.5 mM. The lipids were shaken… to prepare unilamellar vesicles (final volume 590 μL). 70 µL of prepared lipids…
Missing information has been added properly.
- Line 197: why is the PTB signal so noised compared to the WOB one? Measurements of a range of SA-AF532 concentrations (above and below 1 nM) would help to define the Limit of Detection (LoD) and Quantification (LoQ) of PTB.
Response: The coupling efficiency of PTB is not as effective as that of the WOB. This is likely due to the additional polarizing optics present in WOB. We attribute the increased noise observed in PTB to this diminished overall coupling efficiency. We eliminated the polarizing optics in order to simplify the PTB, and enhance its portability. This has been included in the revised manuscript as below.
Revised text (lines 205-207 in the revised manuscript): WOB contains polarizing optics, whereas PTB does not. This is the likely cause of the increased noisiness of the PTB spectra, over the WOB one.
We agree that measurements of a range of concentrations would help to define a limit of detection for PTB. However, given that the purpose of this manuscript was to demonstrate the engineering design of the PTB, and very preliminary functionality via the measurement of streptavidin, we did not perform said measurements. Another reason for this is the very high binding affinity of biotin to streptavidin. Indeed, We propose to perform limit of detection assessments upon application of this biosensor to a biologically relevant analyte. We concur that limit of detection – particularly in a complex biofluid sample – is an incredibly valuable quantity in demonstrating the utility of a sensor for diagnostics applications. In fact, we have previously performed similar characterization many times with WOB for biologically relevant analytes (e.g., Kubicek-Sutherland et al. 2019, Mukundan et al. 2012, etc.). We are currently working on such evaluations for a subsequent, more detailed validation manuscript. We have included a description of this in the conclusion section as below.
Revised text (lines 247-249 in the revised manuscript):We are currently working on validating assay performance, and comparing sensitivity of detection and limit of detection in order to evaluate feasibility for real-world diagnostic applications.
The authors answered the question properly and providing the missing information.
- Line 199-201: the sentence is not clear to me. Each fluorescence acquisition requires a signal normalization in order to exclude any alteration given by the experimental and reagents conditions. Please revise explaining better this statement.
Response: We apologize for the lack of clarity in this sentence. Normalization is traditionally an essential step in the comparison of multiple fluorescence spectra. However, because we acquired spectra in matched pairs by moving our assembled waveguide flow cell between biosensors, normalization is unnecessary. We see that the language we used was unclear, as it was intended to be a general comment on the comparability of fluorimetry assays. We have now replaced the general comment with a specific note on the comparability of these two biosensors.
Old text (lines 199-201 in the initial submission): Because the observed signal of a fluorescence assay will depend on the specific conditions and reagents used in the assay, we instead use the average signal-to-noise ratio (SNR)…
Revised text (lines 208-211 in the revised manuscript): Because PTB and WOB use different spectrometers with different scales of relative fluorescent intensity, we instead use the average signal-to-noise ratio (SNR)…
The text has been modified in a clearer version.
- Line 213: which concentration of SA-AF532 has been used to perform the test reported in Figure 3? Please include this specification.
Response: The caption of Figure 3 has been updated to include the concentration of SA-AF532 (1 nM) used to acquire the spectrum.
Old text (lines 213-214 in the initial submission): Graphical representation of signal (blue) and noise (red), as used to estimate the signal-to-noise ratio (SNR) in this work.
Revised text (lines 223-225 in the revised manuscript): Graphical representation of signal (blue) and noise (red) on a spectrum of PTB recorded with 1 nM SA-AF532, as used to estimate the signal-to-noise ratio (SNR) in this work.
The missing information has been included accordingly to my suggestion.
- Line 220-222: literature reports evidences of stable and enhanced fluorophores and surface functionalizations for highly sensitive optical biosensing applications that here are not mentioned: "Jeong Y, Kook YM, Lee K, Koh WG. Metal enhanced fluorescence (MEF) for biosensors: General approaches and a review of recent developments. Biosens Bioelectron. 2018 Jul 15;111:102-116. doi: 10.1016/j.bios.2018.04.007. Epub 2018 Apr 7. PMID: 29660581"; "L. Sciuto, M.F. Santangelo, G. Villaggio, F. Sinatra, C. Bongiorno, G. Nicotra, S. Libertino, Photo-physical characterization of fluorophore Ru(bpy)32+ for optical biosensing applications, Sensing and Bio-Sensing Research 6, 2015, 67-71" ; "Mukundan H, Anderson AS, Grace WK, Grace KM, Hartman N, Martinez JS, Swanson BI. Waveguide-based biosensors for pathogen detection. Sensors (Basel). 2009;9(7):5783-809"; "Sciuto EL, Bongiorno C, Scandurra A, Petralia S, Cosentino T, Conoci S, Sinatra F, Libertino S. Functionalization of Bulk SiO2 Surface with Biomolecules for Sensing Applications: Structural and Functional Characterizations. Chemosensors. 2018; 6(4):59". I would include these evidences, highlighting the importance of using such stable dyes and efficient surface modifications in order to avoid SNR, sensitivity and specificity common issues of optical detection.
Response: These references have been added where specified, along with the context that other enhanced fluorophores and surface functionalization techniques could be used to further improve the sensitivity of this biosensing platform.
Old text (lines 220-224 in the initial submission): Therefore, PTB is more than sufficient for use with our biosensing assays, which generally employ bright fluorophores with high excitation efficiencies and quantum yields [27-32], and the advantage of the portability of this platform outweighs the disadvantage of reduced sensitivity.
Revised text (lines 230-236 in the revised manuscript): Therefore, PTB is more than sufficient for use with our biosensing assays, which generally employ bright fluorophores with high excitation efficiencies and quantum yields [29-34]. Additionally, further optimizations such as metal-enhanced fluorophores and other surface functionalizations could be employed in the future to improve the sensitivity of this biosensing platform [10,45-47]. Ultimately, the advantage of the portability of this platform outweighs the disadvantage of reduced sensitivity.
A deeper analysis of related literature has been added increasing the importance of further advancements.
- A complete evaluation of the platform analytical performances is missing. In my opinion, going deeper in detail with the experiments, reporting a complete characterization of the detection module (PTB/PEGASUS), a sensing calibration at various concentration of strepatividin-biotinylated lipid bilayer complex and the relative LoD/LoQ extimation, is necessary to prove the sensor suitability for PoC applications.
Response: The purpose of this first concise communication was intended to demonstrate the engineering features, and preliminary functionality of the biosensor platform. The biosensor’s hardware is not a limiting factor in the analytical sensitivity of the sensor. In addition, we have ensured that the sensing components of the well-validated WOB are not compromised or altered in the engineering design of the PTB, which is what we present in this first short paper. We will be following this up with a full article on comparative assay validation and associated metrics. In fact, we have been working on proteins, nucleic acid, and lipid detection on this interface, and this will be covered in future articles.
- A direct sensing tests with the PEGASUS platform, and not its test-bed analogue, would increase the strength of work, especially using more significant molecular targets for on-site diagnostics than streptavidin/biotin complex.
Response: We agree that a direct comparison between PEGASUS and WOB would be ideal rather than using PTB as an analogue of PEGASUS. However, as we noted, the waveguide mounting systems used by PEGASUS and WOB are not interchangeable. This would bias the initial comparisons, which is why we developed the PTB platform. The comparison between biosensors that we present here is only valid because the waveguide, flow cell, lipids, etc. were identical and moved between biosensors. We propose to introduce the actual PEGASUS platform, and perform a three way comparison in the next validation manuscript.
I agree that this is just a concise communication and that future works will address the missing technological sensing characterization of PEGASUS towards all biological targets mentioned. Therefore, some statements need to be revised as follow:
Line 192-193: “PEGASUS and WOB use the same assay architectures”. The sentence needs to be changed using the term “similar” instead of “same” since, as the same authors said, the waveguide mounting systems used by PEGASUS and WOB are not interchangeable. Moreover, the additional polarizing optics present in WOB is missing in PTB/PEGASUS, that is the reason of analytical noise during measurements.
Line 195-197: "through the direct comparison between PTB and WOB presented in this manuscript, thesensing potential of PEGASUS can be inferred for biologically relevant targets, such as those that we have previously assayed". That's not true, in my opinion. No direct comparison between PEGASUS and WOB, proving the effectiveness of proposed portable sensing system, has been done yet. Moreover, a sensing potential that has been tested with streptavidine/biotin cannot be inferred working with other biological targets. Passing from a benchtop to a portable system, a lot of factors can influence the sensing potential depending on the type of biomolecule investigated. As an exmaple: the stabilty of such sensing elements as enzymes and RNA is lower than DNA; the thermal control in the nucleic acids hybridization has to be finely managed since also 1 °C can affect the quality of reaction; the complexity of some biological matrix as blood or urine can increase the noise of measurements. If authors don't have data from others biosensing applications, I suggest to remove the statement.
Lastly, there is still a minor revision that has to be done:
Line 241: "Portable biosensors enable rapid, on-site analytical measurements". That's true for the on-site measurements but not for the speed of analysis, that is given by the integration and simplification of the sensing technology in PoC. I suggest to revise as follow: "Integrated and portable biosensors in a PoC format enable rapid and on-site analytical measurements".
Author Response
Response to Reviewers, Round 2 – biosensors-1614941
Kocheril et al. – Portable waveguide-based optical biosensor
We again thank the reviewers for their thorough review of our manuscript and thoughtful feedback. Our responses to each item are specified below:
Reviewer 1
Comment: Line 192-193: “PEGASUS and WOB use the same assay architectures”. The sentence needs to be changed using the term “similar” instead of “same” since, as the same authors said, the waveguide mounting systems used by PEGASUS and WOB are not interchangeable. Moreover, the additional polarizing optics present in WOB is missing in PTB/PEGASUS, that is the reason of analytical noise during measurements.
Response: We again thank the reviewer for their feedback. By “the same assay architectures,” we referred to the biological assay architecture based on the supported lipid bilayer on the surface of the waveguide rather than the optics and hardware used in the sensor. But we realize that this description may be confusing and therefore we have now clarified this distinction and substituted “the same” for “similar” as suggested by the reviewer.
Old text (lines 192-193): Because PEGASUS and WOB use the same assay architectures…
Revised text (lines 192-193): Because PEGASUS and WOB use similar biological assay architectures…
Comment: Line 195-197: "through the direct comparison between PTB and WOB presented in this manuscript, the sensing potential of PEGASUS can be inferred for biologically relevant targets, such as those that we have previously assayed". That's not true, in my opinion. No direct comparison between PEGASUS and WOB, proving the effectiveness of proposed portable sensing system, has been done yet. Moreover, a sensing potential that has been tested with streptavidine/biotin cannot be inferred working with other biological targets. Passing from a benchtop to a portable system, a lot of factors can influence the sensing potential depending on the type of biomolecule investigated. As an example: the stability of such sensing elements as enzymes and RNA is lower than DNA; the thermal control in the nucleic acids hybridization has to be finely managed since also 1 °C can affect the quality of reaction; the complexity of some biological matrix as blood or urine can increase the noise of measurements. If authors don't have data from others biosensing applications, I suggest to remove the statement.
Response: We thank the reviewer for pointing out that the sensing potential of PEGASUS in the field will likely differ significantly from that of PEGASUS in a laboratory environment, given the sensitive nature of biological assays. We have removed this statement as suggested by the reviewer.
Comment: Line 241: "Portable biosensors enable rapid, on-site analytical measurements". That's true for the on-site measurements but not for the speed of analysis, that is given by the integration and simplification of the sensing technology in PoC. I suggest to revise as follow: "Integrated and portable biosensors in a PoC format enable rapid and on-site analytical measurements".
Response: We have updated our language as suggested by the reviewer.
Old text (line 241): Portable biosensors enable rapid and on-site analytical measurements…
Revised text (line 241): Integrated and portable biosensors in a point-of-care format enable rapid and on-site analytical measurements…
Reviewer 3 Report
The authors have improved the manuscript, issues have been elucidated, and I can recommend its publication in Biosensors.
Author Response
Thank you for your acceptance of our edits to the manuscript.
Sincerely
Harshini Mukundan, Corresponding author